# Multi-Omics Approaches Provide New Insights into the Identification of Putative Fungal Effectors from *Valsa mali*

**DOI:** 10.3390/microorganisms12040655

**Published:** 2024-03-26

**Authors:** Gulnaz Kahar, Yakupjan Haxim, Abdul Waheed, Tohir A. Bozorov, Xiaojie Liu, Xuejing Wen, Mingqi Zhao, Daoyuan Zhang

**Affiliations:** 1State Key Laboratory of Desert and Oasis Ecology, Key Laboratory of Ecological Safety and Sustainable Development in Arid Lands, Xinjiang Institute of Ecology and Geography, Chinese Academy of Sciences, Urumqi 830011, China; gulinazikahaer20@mails.ucas.ac.cn (G.K.); yakoo@ms.xjb.ac.cn (Y.H.); drwaheed@ms.xjb.ac.cn (A.W.); liuxiaojie@ms.xjb.ac.cn (X.L.); wenxj@ms.xjb.ac.cn (X.W.); zhaomingqi19@mails.ucas.ac.cn (M.Z.); 2Xinjiang Key Laboratory of Conservation and Utilization of Plant Gene Resources, Xinjiang Institute of Ecology and Geography, Chinese Academy of Sciences, Urumqi 830011, China; 3Turpan Eremophytes Botanical Garden, Chinese Academy of Sciences, Turpan 838008, China; 4University of Chinese Academy of Sciences, Beijing 100049, China; 5Laboratory of Molecular and Biochemical Genetics, Institute of Genetics and Plants Experimental Biology, Uzbek Academy of Sciences, Yukori-Yuz, Kibray 111226, Tashkent Region, Uzbekistan

**Keywords:** fungal effectors, *Malus sieversii*, RNA-seq, *Valsa mali*

## Abstract

Pathogenic fungi secrete numerous effectors into host cells to manipulate plants’ defense mechanisms. *Valsa mali*, a necrotrophic fungus, severely impacts apple production in China due to the occurrence of Valsa canker. Here, we predicted 210 candidate effector protein (CEP)-encoding genes from *V. mali*. The transcriptome analysis revealed that 146 CEP-encoding genes were differentially expressed during the infection of the host, *Malus sieversii*. Proteome analysis showed that 27 CEPs were differentially regulated during the infection stages. Overall, 25 of the 146 differentially expressed CEP-encoding genes were randomly selected to be transiently expressed in *Nicotiana benthamiana*. Pathogenicity analysis showed that the transient expression of VM1G-05058 suppressed BAX-triggered cell death while the expression of VM1G-10148 and VM1G-00140 caused cell death in *N. benthamiana*. In conclusion, by using multi-omics analysis, we identified potential effector candidates for further evaluation in vivo. Our results will provide new insights into the investigation of virulent mechanisms of *V. mali*.

## 1. Introduction

In general, plants have evolved two strategies of defense mechanisms against fungal pathogens. In the first line of defense, conserved microbial features are recognized as pathogen-associated molecular patterns (PAMPs), triggering the immune response known as PAMP-triggered immunity (PTI) [1]. To overcome this immunity, fungi produce effectors, which can be either harmful secondary compounds or proteins. However, plants have evolved resistance (R) proteins capable of recognizing these fungal effectors, leading to the activation of a second layer of defense known as effector-triggered immunity (ETI) [2].

Fungal effectors are typically secreted proteins that are rich in cysteine. Effector proteins are generally categorized as extracellular effectors, secreted into the apoplasts or xylem of the host, and cytoplasmic effectors, which are delivered into the host cells [2]. Scientific evidence indicates that fungi exert their effectors in two primary sites: within host cells and within the apoplastic space separating neighboring cells [3]. However, the mechanism by which effector proteins are transported to host cells remains unknown. Proteases from plants and fungi additionally process the extracellular effectors at the N- and occasionally C-terminus. Another shared characteristic among extracellular effectors and effectors within host cells is the occurrence of multiple cysteine residues. These cysteine residues potentially participate in the formation of disulfide bridges, enhancing protein stability within the challenging protease-rich environment of the host apoplast [4].

During pathogen infection, salicylic acid (SA) and jasmonic acid (JA) play crucial roles as plant hormones, often exhibiting antagonistic effects on plant defense responses. The maize pathogen *Ustilago maydis* secretes Cmu1, a chorismate mutase, into the host cell to decrease the levels of SA [5]. In addition, when expressed within plant cells, the effectors Pslsc1 and Vdlsc1 reduce SA levels by helping *Phytophthora sojae* maintain full virulence, while Vdlsc1 and Pslsc1 help *Verticillium dahliae* maintain full virulence [6]. During pathogenesis, plant pathogenic nematodes have demonstrated the secretion of chorismate mutases into plant cells. This observation suggests that the manipulation of SA levels serves as a shared objective among numerous plants’ pathogenic organisms. JA signaling has also been shown to be affected by effectors. It is essential for the establishment of mutualism between *Laccaria bicolor* and plants that the ectomycorrhizal fungus secretes the effector MiSSP7 during plant colonization. The transcriptional repressor protein PtJAZ6 is interacted with and prevented from being degraded by MiSSP7 to prevent DNA synthesis alterations caused by JA [7].

Effector candidates can be predicted bioinformatically by determining the presence of a secretion signal at the N-terminus. The expression profile of effector encoding genes changes during different infection stages depending on the type of cell and/or organ being infected [8]. There has been a surge in data related to the identification of putative (secreted) effectors through whole-genome sequencing of fungal pathogens. Integrating genetic, transcriptomic, proteomic, and metabolomic data can enhance the search for effector candidates. A comparative analysis of secretomes and the conducting of BLAST sequence similarity searches can also serve as effective tools for identifying potential effectors within sequenced genomes [3]. For instance, the genome of *Laccaria bicolor* has several MiSSP effectors which are homologous to effectors in rust and other fungi [7,9]. EffectorP 3.0 is a useful tool for predicting effectors in both fungi and oomycetes, capable of distinguishing between apoplastic and cytoplasmic effectors [10]. Additionally, SignalP 6.0, a machine learning model, can accurately detect all five types of signal peptides (SPs) in metagenomic data [11]. EffHunter is an easy and fast bioinformatics tool for the identification of effectors [12]. Moreover, a recent investigation employed AlphaFold2, an AI system that forecasts the 3D configuration of a protein, to mimic the arrangement of 26,653 secreted proteins of 14 significant fungal pathogens relevant to agriculture, along with 6 non-pathogenic fungi and 1 oomycete [13].

*Valsa mali*, the necrotrophic fungus known as apple canker, leads to significant necrosis in apple trees. The disease causes severe yield losses each year in Eastern Asia, where it is the most devastating pathogen of apples [14]. *V. mali* infects woody plants such as pears, crabapples, apricots, peaches, and cherry trees, but it preferentially infects apple trees [15]. Previous studies have shown that the effector protein of *V. mali* plays a very important role in the process of infecting its host, and several effector proteins have been reported [16,17]. For example, the effector protein VmEP1 interacts with host MdPR10 protein, ultimately inhibiting the resistance mediated by MdPR10 against *V. mali* [18]. Another effector, Vm1G-1794, protects the aggregated MdEF-Tu from autophagic degradation to promote infection in apple [19].

Although there have been scattered reports on *V. mali* effectors in the early stages, there is no comprehensive and systematic identification method for its effector proteins. Here, we identified *V. mali* effector-protein-coding genes at the whole-genome level and ultimately confirmed 210 effector proteins through a combination of transcriptome and proteome data, as well as experimental verification in plants. Our study provides valuable data for studying the molecular mechanisms of Vmali infection and a method for verifying the function of *V. mali* effector proteins in plants.

## 2. Material and Methods

### 2.1. Fungal Growth and Infection

The wild-type strain of *V. mali* (EGI-1) from our earlier study [20] was cultured on potato dextrose agar (PDA, 20% potato, 2% glucose, 1.5% agar) medium at 25 °C in the dark.

A protocol with minor modifications was used to surface-sterilize cut-off Malus sieversii twigs grown in the greenhouse and inoculated with fungi as described by Liu and Li [21]. In summary, we first rinsed the healthy twig segments with sterile ddH2O, and then immersed them in 70% ethanol for a duration of 10 minutes. Subsequently, another round of rinsing with sterile ddH2O was performed. The twigs were punctured using a sterilized fabric pattern wheel. Additionally, we inoculated the twigs with a mycelial plug measuring 5 mm, which was aseptically excised from the edge of a 5-day-old *V. mali* colony. EGI1 was cultured on PDA medium. All of the inoculated twigs were incubated at 25 ℃ in dark conditions with high humidity (90% RH) for 5 days.

### 2.2. Genome-Wide Identification of Effector Candidates from V. mali

For the prediction of effector candidates, the protein sequences of *V. mali* were retrieved from the NCBI database (https://www.ncbi.nlm.nih.gov/protein/, accessed on 18 February 2023). SignalP 6.0 was used to search each protein's signal peptide sequences and transmembrane domains (https://services.healthtech.dtu.dk/services/SignalP-6.0/, accessed on 25 February 2023) and DeepTMHMM 1.0.24 (https://dtu.biolib.com/DeepTMHMM, accessed on 10 March 2023) separately. Protein lengths and cysteine residues were calculated for each protein using Novopro online tools (https://www.novopro.cn/tools/, accessed on 20 March 2023).

### 2.3. Transcriptome Sequencing and Data Analysis

Three bark samples with fungal mycelium were collected at 1, 2, and 5 days post-inoculation for transcriptome sequencing. Fungal mycelium samples grown on PDA medium for 5 days were used as controls. We performed transcriptome sequencing as described in a previous article [21]. All raw reads were deposited in the Sequence Read Archive (SRA) database of the National Center for Biotechnology Information. (NCBI: https://www.ncbi.nlm.nih.gov/, accessed on 25 March 2023) under project accession number PRJNA687214 [21].

To carry out transcriptome data analysis, we obtained the gene model annotation files and reference genome pertaining to *V. mali* from NCBI (BioProject: PRJNA268126) directly. An index of the reference genome was built using Hisat2 (v2.1.0). Transcriptome data (NCBI BioProject: PRJNA687214) were used and paired-end clean reads were aligned to the reference genome of *V. mali* by using Hisat2. In this study, HTSeq v0.6.1 was utilized to determine the count of reads that were mapped to every gene. By taking into account the gene’s length and the number of reads mapped to it, the FPKM value for each gene was calculated. This calculation allows for the normalization of gene expression levels, taking into consideration both the length of the gene and the total number of mapped reads. Differential expression analysis was conducted on two groups using the ‘DESeq’ R package (1.18.0). The Benjamini–Hochberg approach was used to adjust the P-values and control the false discovery rate. Transcripts with an adjusted P-value 1 were deemed significantly differentially expressed. The generated heatmap was based on the log2 fold-change values. The differential expression analysis of two groups was performed by utilizing the ‘DESeq’ R package (1.18.0). To control the false discovery rate, the *p*-values were adjusted using the Benjamini–Hochberg approach. Significantly differentially expressed transcripts were identified based on an adjusted *p*-value 1. The generation of the heatmap involved the utilization of log2 fold-change values.

### 2.4. Proteome Sequencing and Data Analysis

Three independent bark samples were collected 1, 2, and 5 days post-inoculation (dpi) for proteome sequencing. Fungal mycelia samples (0 dpi) on PDA medium were used as controls. Each sample was individually ground using liquid nitrogen and then subjected to lysis using SDT lysis buffer (consisting of 4% SDS, 100 mM DTT, and 10 mM TEAB). To further enhance the lysis efficiency, the samples were subjected to ultra-sonication in ice water for 5 min. Subsequently, the lysate was subjected to thermal treatment at 95 °C for 8 min, followed by centrifugation at 12,000× *g* for 15 min at 4 °C. The resulting supernatant was then mixed with 10 mM DTT and incubated at 56 °C for 1 h. To ensure complete alkylation, the lysate was treated with adequate iodoacetamide for 1 h at room temperature. The samples were mixed with four times the volume of pre-cooled acetone by vortexing, followed by incubation at a temperature of −20 °C for a minimum duration of two hours. Subsequently, the samples underwent centrifugation at 12,000× *g* for a period of 15 min, utilizing a temperature of 4 °C, and pellets were collected. After washing with 1 mL cold acetone, the pellet was dissolved with dissolution buffer (8 M Urea, 100 mM TEAB, pH 8.5). According to the manufacturer’s instructions, total proteins were quantified using the Bradford Protein Assay Kit (P0006, Beyotime, Shanghai, China).

To dissolve each protein sample, DB dissolution buffer (8 M Urea, 100 mM TEAB, pH 8.5) was used to prepare a volume of 100 µL. The sample was mixed with trypsin and 100 mM TEAB buffer and then digested at 37 °C for 4 h. The trypsin and CaCl_2_ were used to digest the sample overnight. In order to achieve a pH of approximately 3, formic acid was added and centrifuged for 5 min at room temperature with a speed of 12,000× *g*. The resulting supernatant was then carefully loaded into the C18 desalting column (Waters, Milford, MA, USA, BEH C18, 4.6 × 250 mm, 5 mm). The column was washed three times using a buffer solution consisting of 0.1% formic acid and 3% acetonitrile. The elution of the desired components was carried out by employing a buffer solution containing 0.1 formic acid and 70% acetonitrile. Each sample’s eluent was collected and lyophilized. It was reconstituted with 100 µL of 0.1 M TEAB buffer, and 41 µL of acetonitrile-dissolved TMT labeling reagent (Thermo Fisher Scientific, Shanghai, China) was added, and the sample was shaken for 2 h at room temperature. The reaction was terminated by adding 8% ammonia. The labeled samples were then mixed in equal volumes, and desalinating, and lyophilizing were then performed. To carry out a gradient elution, a mobile phase A consisting of 2% acetonitrile and with pH adjusted to 10.0 using ammonium hydroxide, along with a mobile phase B containing 98% acetonitrile, was utilized. Initially, the lyophilized powder was dissolved in solution A, followed by centrifugation at room temperature (~25 °C) and 12,000× *g* for a duration of 10 min. Next, the resulting supernatant was subjected to fractionation using a C18 column (Waters, USA) with dimensions of 4.6 × 250 mm and a particle size of 5 μm, integrated within a Rigol L3000 high-performance liquid chromatography (HPLC) system. The eluates were continuously examined at a UV wavelength of 214 nm, with the column oven maintained at a constant temperature of 45 °C. The eluate was collected into separate tubes every minute, ultimately forming 10 fractions. Once all the fractions had been completely dried under vacuum, they were reconstituted in water containing 0.1% formic acid (FA).

We carried out shotgun proteomic investigations employing a Q ExactiveTM HF-X mass spectrometer (Thermo Fisher Scientific, China) in the data-dependent acquisition (DDA) mode, supported by an EASY-nLCTM 1200 UHPLC system (Thermo Fisher Scientific, China), to establish a transition library. A sample of 1 µg was injected into a C18 Nano-Trap column (4.5 cm by 75 cm by 3 m). Peptides were separated into analytical columns (15 cm × 150 μm, 1.9 μm). Using a Q ExactiveTM HF-X mass spectrometer from Thermo Fisher Scientific, China, peptides were separated and analyzed with an ion source from Nanospray FlexTM (ESI), a spray voltage of 2.3 kV and a capillary temperature of 320 °C. A full range scan was conducted, covering the *m*/*z* range from 350 to 1500, utilizing a resolution of 60,000 (at *m*/*z* 200). To maintain control, the automatic gain control (AGC) target value was set at 3 × 10^6^, while the maximum ion injection time allowed was 20 ms. The top 40 precursors with the highest abundance in the full scan were chosen for fragmentation through high energy collisional dissociation (HCD). The resulting fragments were then subjected to MS/MS analysis, employing a resolution of 45,000 (at *m*/*z* 200) for 10 plex. For this stage, the automatic gain control (AGC) target value was adjusted to 5 × 10^4^, and the maximum ion injection time was extended to 86 ms. Additionally, a normalized collision energy of 32% was applied, coupled with an intensity threshold of 1.2 × 10^5^. Finally, the dynamic exclusion parameter was established to be 20 s.

The resulting spectra from each run were searched separately against the *V. mali* protein database (https://www.ncbi.nlm.nih.gov/protein/?term=Valsa-mali, accessed on 18 February 2023) using the following search engine: Proteome Discoverer 2.2 (PD 2.2, Thermo). The parameters searched were set as follows: mass tolerance for precursor ion was 10 ppm and mass tolerance for product ion was 0.02 Da. The carbamidomethyl modification was specified as fixed, whereas the oxidation of methionine (M) and the TMT plex were specified as dynamic modifications, and the acetylation and TMT plex were specified as N-terminal modifications. A maximum of 2 missed cleavage sites were allowed. A further filter was applied to the retrieval results as part of software PD 2.2 to improve the accuracy of the analysis results: peptide spectrum matches (PSMs) with credibility over 99% were identified as true PSMs. At least one unique peptide is present in the identified protein. The identified PSMs and protein were retained and performed with FDR no more than 1.0%. The protein quantitation results were statistically analyzed using a *T*-test. The proteins whose quantitation was significantly different between experimental and control groups (*p* < 0.05 and |log2 foldchange| > 0) were characterized as differentially expressed proteins (DEP).

### 2.5. Plant Growth, Vector Construction

To express transient genes, *Nicotiana benthamiana* plants were grown at 25 °C with 60% humidity under photoperiodic lighting: 16/8 h light/dark. To conduct molecular cloning, a strain of Escherichia coli, specifically the Top10 competent cells, was utilized. Additionally, Agrobacterium tumefaciens strain GV3101, which was obtained from TransGen Biotech in Beijing, China, was employed for transient gene expression.

RNA was extracted from the *V. mali* strain using the Takara MiniBEST Universal RNA Extraction Kit (Takara, Beijing, China). The reverse transcription of cDNA from *V. mali* RNA was performed using the TAKARA PrimeScriptTM RT reagent Kit (Perfect Real Time Kit, Takara, Beijing, China) with gDNA Eraser. Using 2× Phanta Max Master Mix, we performed PCR to clone the CDS sequence of *V. mali* target effector encoding genes (Vazyme, Nanjing, China), and sequence-specific primers (Appendix A) were used. These target genes subsequently cloned into the pGR107 plant expression vector. The PCR products and pGR107 vectors were double-digested with recognition sites at the 5′-(Cla I) and 3′-termini (Sal I) restriction enzymes, and ligated with T4 ligase (Takara, Beijing, China). The resulting plasmid sequence was confirmed through the use of PCR and DNA sequencing by Sangon Biotech. (Shanghai, China).

### 2.6. In Planta Expression and Symptom Analysis

In this study, the researchers used pGR107 vectors to express the mouse cell death inducer BAX [22], and GFP alone was derived from previous studies and employed as the control [23]. Additionally, they utilized A. tumefaciens with pGR107 vectors containing *V. mali* effector candidate proteins. To introduce these vectors into *N. benthamiana* leaves, they followed the infiltration protocol described by Oh and Young [24]. After 24 h, *N. benthamiana* leaves that had been infiltrated previously were subjected to infiltration again using a specific *A. tumefaciens* strain containing a vector that was harboring the BAX gene. For experimental purposes, *A. tumefaciens* cells carrying the pGR107-BAX vector were considered positive controls, while those carrying the empty vector pGR107-GFP served as negative controls. Each assay utilized a total of at least 30 leaves sourced from ten distinct plants. Subsequent to Trypan blue staining, the lesion area was evaluated through ImageJ 1.52a;1.8.0_112 software [25].

### 2.7. Validation of the SP Secretory Activities of Candidate Effectors

The yeast strain YTK12 and the pSUC2 vector used for the validation of predicted signal peptides were kindly provided by Prof. Wei Yao (State Key Lab for Conservation and Utilization of Subtropical Agri-Biological Resources, Guangxi Key Lab of Sugarcane Biology, Guangxi University).

The yeast secretion assay, as previously documented, was used to conduct the functional validation of predicted SPs for candidate effectors. In order to examine each candidate effector, the primers listed in Appendix A were employed to amplify the predicted SP sequences, which were subsequently inserted into a truncated SUC2 gene encoding an invertase lacking an SP through dual enzyme restriction cutting. The validated constructs were sequenced and subsequently introduced into yeast strain YTK12 lacking invertase activity, employing the Frozen-EZ yeast transformation II kit (Zymo Research, Irvine, CA, USA), as previously described [24]. Transformants were selected using a yeast minimal tryptophan dropout medium, which is composed of 0.67% yeast N base without amino acids, 0.075% tryptophan dropout supplement, 2% sucrose, 0.1% glucose, and 2% agar. To assess invertase secretion, the yeast colonies were replica-plated onto YPRAA plates containing 1% yeast extract, 2% peptone, 2% raffinose, and 2 µg of antimycin A per liter. The invertase secretion signal from Phytophthora sojae Avr1b was used as a positive control, while the N-terminal SP of Mg87 in Pyricularia oryzae served as the negative control.

### 2.8. Statistical Analysis

GraphPad Prism9 was used for data analysis and visualization. For the statistical test, one-way analysis of variance (ANOVA) was used. The level of significance was set at *p* < 0.05.

## 3. Results

### 3.1. Genome-Wide Identification of Candidate Effector Proteins from V. mali

The *V. mali* (isolate 03-8) genome encodes 11284 proteins (amino acids ≥ 50) in total and they were used for the prediction of effector candidates in silico by following multiple steps (Figure 1A). In order to predict candidate effector proteins, we defined CEPs as proteins without transmembrane domains, containing less than 400 amino acids, and containing more than four cysteine residues. Using SignalP 6.0, 806 proteins containing N-terminal SP were identified [26]. Further, 181 proteins were removed by analyzing the transmembrane domain using TMHMM 2.0 (TMHMM score < 1). Based on the peptide size (amino acids × 400) and cysteine residue count (cysteine residues count × 4) exclusion screens, only 210 proteins were found to meet the criteria. These 210 protein sequences were thus considered the final CEPs (Appendix A). The functional annotation revealed that 130 of the 210 predicted CEPs are hypothetical proteins.

All candidate CEPs were distributed on all chromosomes according to genome location analysis (Figure 1B). Of the 210 CEPs, 38.09% (80 sequences) had a size between 300 and 400 amino acids; 3% had a size less than 100 amino acids (Figure 1C). Based on statistical analysis, 89% of the 210 final CEPs contained fewer than 10 cysteines, while 25% contained at least 4 cysteines (Figure 1D). Notably, one CEP (KUI69483.1) contained 40 cysteines.

### 3.2. Transcriptional Analysis of Effector Candidates

To study transcriptional changes in predicted CEP-encoding genes, we analyzed the transcriptome of *V. mali* during the infection of *M.sieversii*. A total of 11,722 genes were differentially expressed (*p* < 0.05, |log2 foldchange| > 1) compared to the control group (Figure 2A), and 1554 genes were common at all the infection stages (Figure 2B). Further analysis showed that 146 CEP-encoding genes were differentially expressed during the infection stage (Figure 2C).

### 3.3. Proteomic Analysis of Effector Candidates

The proteomic analysis was performed on branches of *M. sieversii* infected with *V. mali* after 1, 2, and 5 days post infection. As a control, the *V. mali* fungus was grown on PDA medium. (Figure 3A). Proteome data were generated by Tandem Mass Tags techniques. The proteomic data analysis revealed that 7112 fungal proteins were differentially expressed (*p* < 0.05, |log2 foldchange| > 0) during infection. (Figure 3B,C). Most of the proteins were differentially expressed in the early stages of infection (Figure 3B). In all infection stages, 1919 proteins were found to be common in 1, 2 and 5 dpi (Figure 3C).

The log2 fold-change ratio compared to 0 dpi showed that 27 CEPs were significantly differentially regulated during the infection stages. Apparently, 13 out of 27 differentially expressed CEPs were hypothetical proteins (Figure 3D). Notably, the level of one CEP (KUI65650.1) increased almost twice after infection.

### 3.4. Pathogenicity Analysis of Selected Effector Candidates

The fungi effector is a molecule that mediates the interaction between the fungus and host cell that often triggers cell death or host defense suppression. However, necrotrophic fungal effectors often induce cell death in the host plant as part of their virulence [27]. In order to examine the function of the anticipated CEPs, a random selection was made of 25 CEPs that displayed differential expression through transcriptomic data analysis. The capacity to initiate cell death or inhibit BAX-induced cell death in *N. benthamiana* was then tested. The outcomes revealed that 22 out of the 25 chosen CEPs did not demonstrate either cell death induction or the suppression of BAX-induced cell death (Figure 4). In contrast, two CEPs significantly induced cell death in *N. benthamiana* (Figure 5A,B), while one CEP suppressed BAX-induced cell death (Figure 5C). Of these three CEPs, KUI74483.1 (VM1G-10148) is the PRY2 protein, KUI64965.1 (VM1G_00140) is a hypothetical protein that has homologs in the GenBank NR database, and KUI69830.1 (VM1G-05058) is putative pectate lyase A.

### 3.5. Secretion Analysis of Selected Effector Candidates

To confirm the secretion of selected CEPs, we used an invertase secretion assay in yeast mutant YTK12 cells [28]. In the vector pSUC2, the signal peptide sequences of three effector candidate genes, VM1G-10148, VM1G-00140, and VM1G-05058, were fused in frame to the mature sequence of yeast invertase [29]. The potential secretory functions of all three candidate effector proteins were indicated by the N-terminal signal peptides they possessed. To serve as a negative control, strains that underwent transformation with the pSUC2 vector exhibited growth solely on CMD-W, while no growth was observed on the YPRAA medium. Conversely, strains harboring the Avr1b signal peptide, employed as the positive control, displayed growth on both CMD-W and YPRAA media. The integration of five effector candidate gene constructs facilitated the growth of the invertase mutant yeast strain YTK12 on YPRAA medium (with raffinose instead of sucrose, and growth only when invertase was secreted) (Figure 6). These results suggested that these two proteins (VM1G-10148, VM1G-05058) were typical secretory proteins; VM1G-00140 may have unique secretory pathways.

## 4. Discussion

The role of effector proteins in the pathogenic fungal invasion and colonization of plant hosts is crucial [30]. Based on the *V. mali* genome, we predicted 210 CEPs. In total, 42 of 210 CEPs contained at least 10 cysteines, and 1 CEP(KUI69483.1) contained 40 cysteines. The cysteine-rich apoplastic effectors (CEPs) may function as fungal apoplastic effectors, as it is highly possible that the cysteines found in fungal apoplastic effectors establish intramolecular disulfide bonds. These bonds are crucial for the stability and proper functioning of the apoplast, which is rich in proteases [31].

The functional annotation revealed that 62% (130 CEPs) of the 210 predicted CEPs are hypothetical proteins, suggesting that the existence and function of these proteins needs further evidence. Notably, VM1G_00345 encodes a small protein (92 amino acids) annotated as a hypothetical protein (KUI65650.1), and the protein level was significantly increased during infection, indicating that it plays an important role in host infection. Furthermore, some of the CEPs are cell wall-degrading enzymes (CWDEs), such as endoglucanase, which breaks down cellulose in the host cell wall [32].

Some of the CEPs we predicted were previously reported to demonstrate the reliability of our prediction method. For instance, Vm1g_09394 (KUI73808.1) was reported as a Hec2-domain-containing effector and could lead to cell death when transiently expressed in *N. Benthamian* [33]. Interestingly, some other proteins from *V. mali* were reported as effectors. For example, the transient expression of VM1G_05997(KUI70334.1) in *N. benthamiana* leaves with BAX suggested that VM1G_05997 is an effective cell death suppressor [34]. In our study, VM1G_05997 was omitted since it did not have any cysteine residues. In addition to this, recent studies reported that that the highly induced effector Vm1G-1794 secreted by *V. mali* competes with MdATG8i to inhibit autophagy and deplete MdEF-Tu in the MdATG8i complex. The formation of stable MdEF-Tu aggregates induced by Vm1G-1794 enhances the sensitivity of apple to *V. mali* [19]. The above example shows that our method is likely to have missed very few proteins due to its high screening threshold. The results reinforce the view that functional genomic pipelines can identify effectors particularly well from mined sequences [35,36].

The suppression of BAX-induced cell death has been proven to be a screening tool for fungal effectors [37]. In this study, VM1G-05058 was identified as encoding putative pectate lyase A (KUI69830.1) belonging to CWDE. As a result of further studies, it was revealed that KUI69830.1 could be secreted (Figure 6), and could also suppress BAX-induced cell death (Figure 4 and Figure 5), indicating that it met the criteria for a fungal effector. In order to effectively colonize the host, fungi secrete effector proteins that manipulate host processes [30,38,39]. The prediction method of fungal effector proteins is based on a relatively wide range of criteria, primarily the presence of secretion signals owing to the lack of conservative characteristics [40,41]. Numerous preceding investigations have predominantly focused on discovering the signal peptides of effector proteins [42], employing yeast secretion systems to confirm their secretory functionality [9]. A yeast secretion assay was conducted to confirm the secretion ability of the selected candidate effectors. The signal peptides of VM1G-10148 and VM1G-05058 were demonstrated to be secretory, indicating that these CEPs are secretory proteins. Since VM1G-00140 is not a classical secretory protein, this suggests that other secretory pathways may be involved.

Integrating multiple types of omics data can provide researchers with a better understanding of various aspects of plant-pathogen biology and interactions than studying a single omics type alone [43]. We analyzed the transcriptome and proteome of *V. mali* during the host infection stage in order to further understand CEPs. Our results showed that 157 CEP-encoding genes were differentially expressed during the infection, suggesting that *V. mali* transcriptionally activated their key virulence-related genes to attack the host cell. For instance, VM1G_01234 is a tyrosinase encoding gene and was highly upregulated during the infection. The cytosolic tyrosinases of fungi play a crucial role in the biosynthesis of melanin pigments [44]. A hypothetical protein (KUI72244.1) was encoded by VM1G_07780, and the transcription was upregulated (Figure 2C), as well as the protein level (Figure 3D), during infection. Furthermore, the expression level of the VM1G_08575 gene was upregulated (Figure 2C), while the protein level of Ecm33 (KUI72888.1), encoded by VM1G_08575, was significantly decreased (Figure 3D). Through regulating the integrity of fungal cell walls, ECM33, a glycosylphosphatidylinositol (GPI)-anchored protein, is important for fungal development and infection [45]. The Ecm33 mutant showed reduced infectivity in maize colonization and produced three- to four-fold fewer conidia in comparison with the wild type and complemented strain. Among other things, Ecm33 is required for the proper composition of cell walls, and it plays an important role in the manufacture of aflatoxin and in the colonization of seeds by *Aspergillus flavus* [46].

In summary, in this study we predicted 210 candidate effectors from *V. mali*. Through a combination of multiomics data, 25 CEPs were tested to induce cell death or suppress BAX-induced cell death, and 1 CEP was identified to secrete and suppress BAX-induced cell death. Our results provide an innovative approach to identifying fungal effector proteins to better understand the mechanisms of infection of *V. mali*. 

## Figures and Tables

**Figure 1 microorganisms-12-00655-f001:**
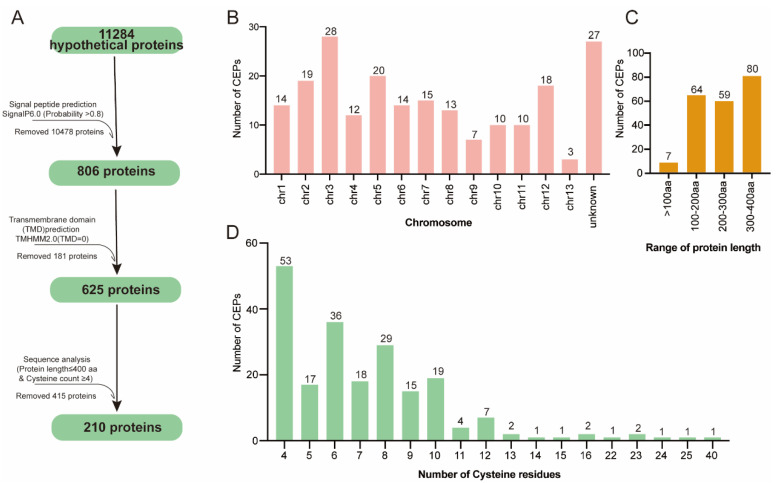
Prediction of CEP-encoding genes from the *V. mali* genome. (**A**) Genome-wide prediction pipeline for CEPs. (**B**) Chromosomal distributions of CEP-encoding genes. (**C**) Distribution of CEPs by length and (**D**) by cysteine residues.

**Figure 2 microorganisms-12-00655-f002:**
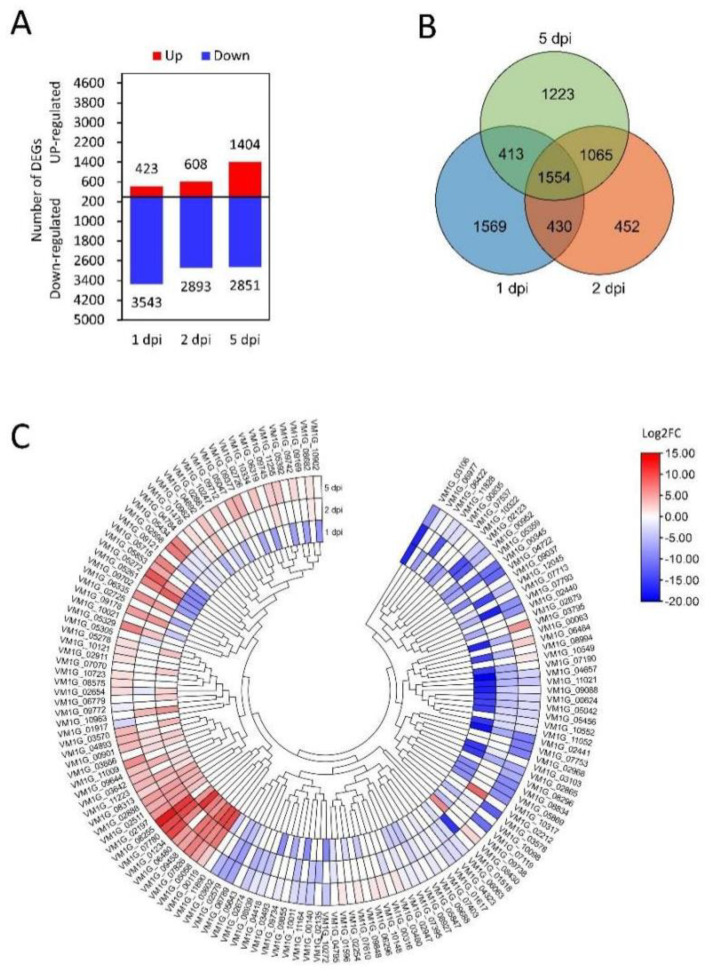
Expression profiles of *V. mali* genes during the infection of *M. sieversii*. (**A**) Amount of differential gene expression. (**B**) Venn diagram of the differentially expressed genes at the different time points. (**C**) Expression heat map of differentially expressed CEP-encoding genes. Each panel shows a single sample, and each cell shows the relative expression level for a single gene. Expression levels were indicated by color gradient, with red and blue representing upregulation and downregulation, respectively.

**Figure 3 microorganisms-12-00655-f003:**
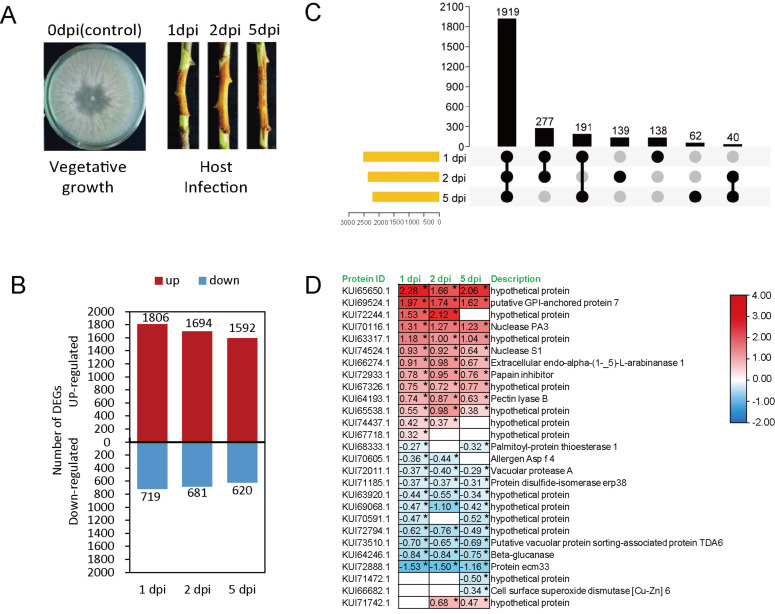
Expression profiles of *V. mali* proteins during the infection of *M. sieversii*. (**A**) Schematic diagram of fungal infection and sampling. Apple bark samples infected with fungal mycelia were collected after 1, 2, and 5 dpi for proteomic analysis. Fungal mycelia on PDA medium used as control (**B**). Number of up- and down-regulated proteins. (**C**) Upset plot of differentially expressed proteins in each sample. The horizontal bars indicate the total number of differentially expressed proteins in each sample and vertical bars indicate the intersection size between sets of differentially expressed proteins within one or more sample(s). Dark connected dots are visual indicators of the intersections. (**D**) Expression heat map of differentially expressed CEPs. Each row shows the relative expression level for a single protein, and each column shows a sample. Expression levels were indicated by color gradient, with red and blue representing upregulation and downregulation, respectively. The number in each cell of the heatmap represents the fold-change expression level. “*” indicates significant differences at *p* < 0.05 according to a one way ANOVA test.

**Figure 4 microorganisms-12-00655-f004:**
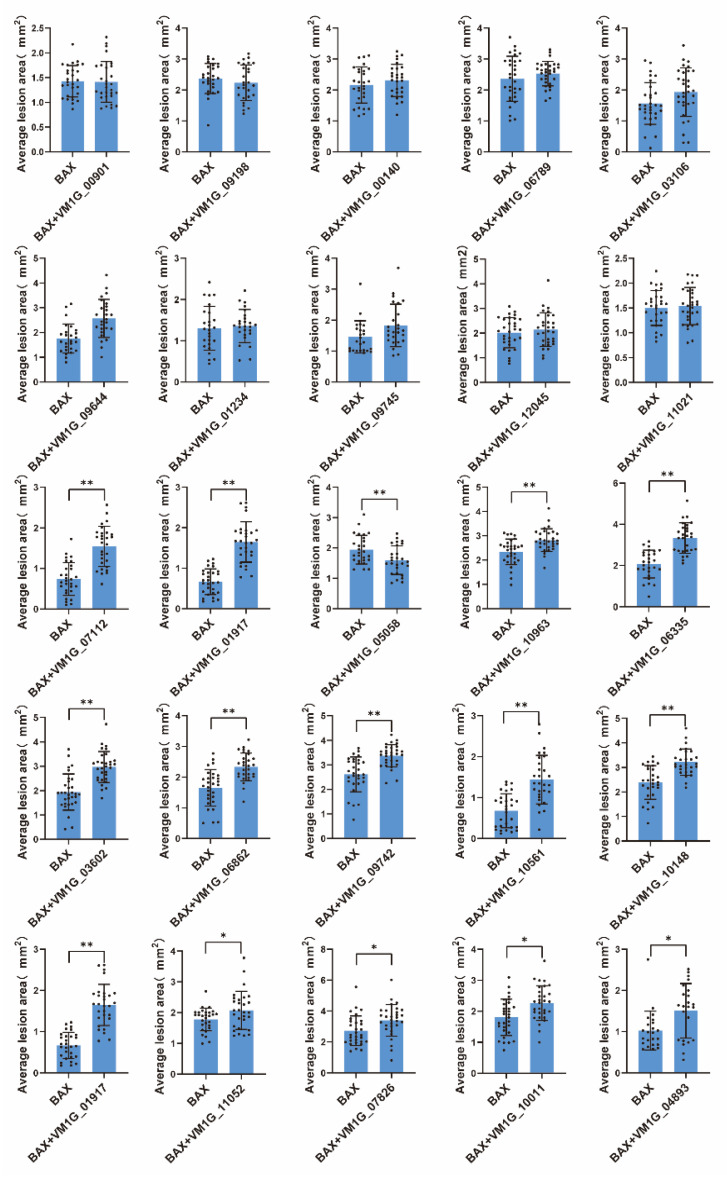
Statistical analysis of symptoms caused by selected CEPs in *N. benthamiana***.** In order to investigate the effects of candidate proteins on *N. benthamiana* leaves, the researchers infiltrated the leaves with *A. tumefaciens* cells containing PVX vectors carrying either green fluorescent protein (GFP) as a negative control or the effector candidate proteins. The lesion area was measured using ImageJ. Error bars represent the mean ± SE of three biological replications. “*” indicates significant differences at *p* < 0.05, “**” indicates significant differences at *p* < 0.01, according to a one way ANOVA test.

**Figure 5 microorganisms-12-00655-f005:**
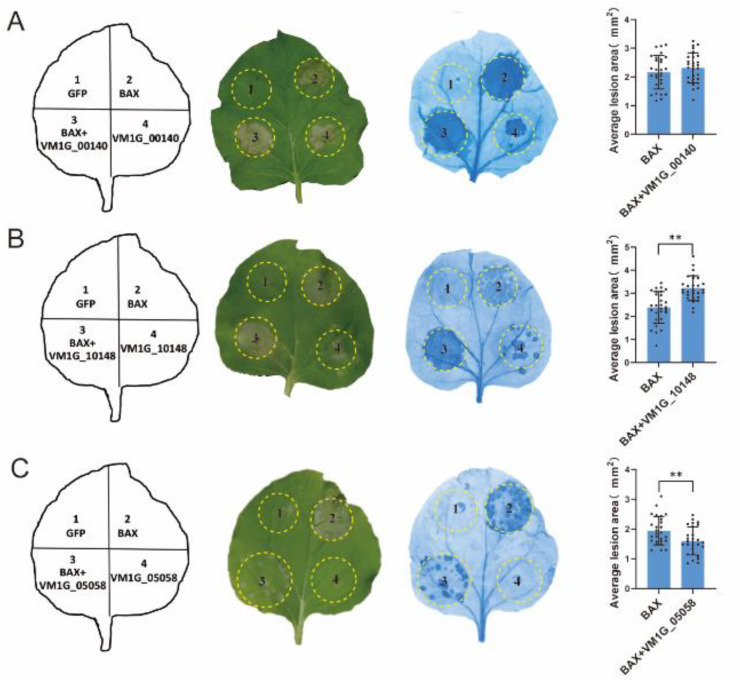
Transient expression of three CEPs in *Nicotiana benthamiana* leaves. Transient expression of VM1G_00140 (**A**), VM1G_10148 (**B**), and VM1G_05058 (**C**). Infiltration of *A. tumefaciens* cells carrying PVX vectors with green fluorescent protein (GFP) (serving as a negative control) or potential effector proteins was performed on *N. benthamiana* leaves. Visualization of cells was facilitated by decolorizing them with ethanol, and photographs were captured after 7 days of infiltration. To ensure reliability, this assay was conducted on a minimum of 30 leaves spread across ten plants. Error bars represent the mean ± SE of three biological replicates. “**” indicates significant differences at *p* < 0.01, according to a one way ANOVA test.

**Figure 6 microorganisms-12-00655-f006:**
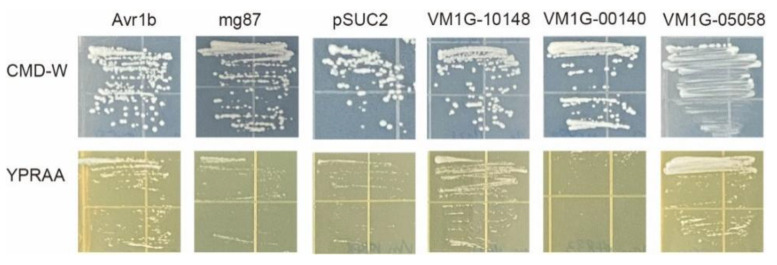
Secretion analysis of signal peptides of selected CEPs. The pSUC2 vector was employed to incorporate the signal peptide, subsequently introducing it into the yeast YTK12 strain. As a positive control, the signal peptide from pSUC2-Avr1b was employed. YTK12 strains carrying the pSUC2 vector and the pSUC2-Mg87 vector were employed as negative controls.

## Data Availability

The transcriptome data used in this study are available from the Sequence Read Archive (SRA) database of the National Center for Biotechnology Information (NCBI: https://www.ncbi.nlm.nih.gov/, accessed on 25 March 2023) under project accession number PRJNA687214 (https://dataview.ncbi.nlm.nih.gov/object/PRJNA687214?reviewer=dve4c2uosci45ag7dnbunbrt44, accessed on 25 March 2023). The proteome data were deposited in the Science Data Bank (https://www.scidb.cn/, accessed on 25 March 2023) under a CSTR ID of 31253.11.sciencedb.11090, and the DOI is https://doi.org/10.57760/sciencedb.11090.

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
