# Peer review of "Multi-Omics Approaches Provide New Insights into the Identification of Putative Fungal Effectors from Valsa mali"

_microorganisms, 2024, doi:10.3390/microorganisms12040655_

Round 1

Reviewer 1 Report

Comments and Suggestions for Authors

This is a technologically sound but rather oddly positioned manuscript that will be of limited value to the broader scientific community. At the top of the Discussion the authors make the following statements; “Fungal effector proteins are predictable, and the methods for prediction of fungal effectors rather mature and software and tools are available (29). Based on 364 V.mali genome, here we predicted 210 proteins as candidate effectors.” If this is indeed so, why the need for this study? And once predicted how and when are these confirmed? This is prceeded by the following statement in the Introduction and is assumed to represent the ‘Aim and objective’ of this study – “Therefore, exploring the full landscape of effector 92 candidate proteins of V. mali (EGI-1 strain) with multi-omics approaches, and validate 93 their pathogenicity during the process of infection to M. sieversii, will provide insight into 94 how M. sieversii resists V. mali infection.”. Thus, in this referees opinion a clearer more directed Aim with a defined subset of objectives is required. As it stands, it is not clear what the work hopes to achieve and/or how this will add to the larger body of knowledge. My second area of concern relates to the description of the Materials and Methods. These are as follows.

What is meant by “healthy Twig segment”? How determined?

“The sterilized twigs” – how sterilized? Describe.

“twigs were incubated at 25 â„ƒ in dark condition with high humidity (90% RH) for 5 days” Why? Physiological basis for this should be described.

What is meant by “fungal mycelial were collected at 1-, 2-, and 5-days post-inoculation”? Collected from where?

“We conducted transcriptome sequencing as described 120 previously.” But no further information is provided. Give a brief overview or provide the necessary References at least.

There are many of these open-ended statements that are either meaningless in the broader context or without foundation. The authors are requested to revisit the text and to improve the quality (scientific language) and its readability so that others can at least attempt tpo corroborate what is being posited.

Comments on the Quality of English Language

Greater care needed in sentence construction and in presentation of information in correct scientific English. Take time to review and to re-read with meaning.

Author Response

1# Reviewer Comments

  1. This is a technologically sound but rather oddly positioned manuscript that will be of limited value to the broader scientific community. At the top of the Discussion the authors make the following statements; “Fungal effector proteins are predictable, and the methods for prediction of fungal effectors rather mature and software and tools are available (29). Based on mali genome, here we predicted 210 proteins as candidate effectors.” If this is indeed so, why the need for this study? And once predicted how and when are these confirmed? This is prceeded by the following statement in the Introduction and is assumed to represent the ‘Aim and objective’ of this study – “Therefore, exploring the full landscape of effector candidate proteins of V. mali (EGI-1 strain) with multi-omics approaches, and validate their pathogenicity during the process of infection to M. sieversii, will provide insight into how M. sieversii resists V. mali infection.”. Thus, in this referee’s opinion a clearer more directed Aim with a defined subset of objectives is required. As it stands, it is not clear what the work hopes to achieve and/or how this will add to the larger body of knowledge.

A: Thank you for question. Fungal effector proteins play a crucial role in fungal infection, and identifying fungal effectors is of great practical significance in understanding the infection mechanism. Although there have been scattered reports on V.mali effectors in the early stage, there is no comprehensive and systematic identification of its effector proteins. In addition, although different prediction tools are used to predict effector proteins, experimental verification is still needed for further confirmation, especially symptom analysis in plants is very necessary. In our study, we identified V.mali effector protein-coding genes at the whole-genome level and ultimately confirmed 210 effector proteins through a combination of transcriptome and proteome data, as well as experimental methods in plants. Our study provides data for studying the molecular mechanism of Vmali infection and a method for verifying the function of V. mali effector proteins in plants.

We also rewrite this part in the manuscript.

  1. My second area of concern relates to the description of the Materials and Methods. These are as follows.

What is meant by “healthy Twig segment”? How determined?

A: Thanks for your question. In this study, “healthy Twig segment” refer to the wild apple branches without wounds, other diseases and not infected by Valsa mali. In order to avoid misleading, we decided to use the word "uninfected twig" in the in the manuscript after consulting the literature.

3.“The sterilized twigs” – how sterilized? Describe.

A: Thanks for your question. In this study, for the sterilization the apple twig segments were rinsed with sterile ddH2O, then immersed in 70% ethanol for 10 minutes. This was repeated by another round of rinsing with sterile ddH2O.

4.“twigs were incubated at 25 ℃ in dark condition with high humidity (90% RH) for 5 days” Why? Physiological basis for this should be described.

A: Thanks for your question. The infection method we used in this study was entirely derived from previous literature reports(1,2,3). According to the literature, the optimal growth temperature range for V.mali is 24℃ to 28℃, and dark treatment and high humidity improve the spore production and mycelial growth. Therefore, for the V.mali infection, we incubated the apple twigs with V.mali under dark conditions at 25℃.

Reference:

  1. Feng H, Xu M, Liu Y, Dong R, Gao X, Huang L (2017) Dicer-Like Genes Are Required for H2O2 and KCl Stress Responses, Pathogenicity and Small RNA Generation in Valsa mali. Frontiers in Microbiology 8 doi:10.3389/fmicb.2017.01166
  2. Yin Z, Ke X, Huang D, Gao X, Voegele RT, Kang Z, Huang L (2013) Validation of reference genes for gene expression analysis in Valsa mali var. mali using real-time quantitative PCR. World Journal of Microbiology and Biotechnology 29(9):1563-1571 doi:10.1007/s11274-013-1320-6
  3. Liu X, Li X, Wen X, Zhang Y, Ding Y, Zhang Y, Gao B, Zhang D (2021) PacBio full-length transcriptome of wild apple (Malus sieversii) provides insights into canker disease dynamic response. BMC Genomics 22(1) doi:10.1186/s12864-021-07366-y

5.What is meant by “fungal mycelial were collected at 1-, 2-, and 5-days post-inoculation”? Collected from where?

A: Thanks for your question. Sampling for transcriptome sequencing described previously (1).

The sterilized twigs were then punctured with a fabric pattern wheel (2 cm in diameter) and inoculated with a mycelial plug (5 mm) excised aseptically from the edge of a 5-day-old V. mali (EGI1), cultured on PDA medium. All of the inoculated twigs were incubated at 25 ℃ in dark condition with high humidity (90% RH) for 5 days. Barks of twigs with canker were harvested at the time points of 0, 1, 2, and 5 dpi and each sample contained three biological replicates. Fungal mycelia samples on PDA medium at 0 days post-inoculation were used as controls.

We also provided more detailed sampling procedures in the manuscript.

Reference

  1. Liu, X., Li, X., Wen, X. et al. PacBio full-length transcriptome of wild apple (Malus sieversii) provides insights into canker disease dynamic response. BMC Genomics 22, 52 (2021). https://doi.org/10.1186/s12864-021-07366-y

6.“We conducted transcriptome sequencing as described previously.” But no further information is provided. Give a brief overview or provide the necessary References at least.

A: Thanks for your question. Transcriptome sequencing described previously (1). We also revised this part and provided more detailed procedures in the manuscript.

Reference

  1. Liu, X., Li, X., Wen, X. et al. PacBio full-length transcriptome of wild apple (Malus sieversii) provides insights into canker disease dynamic response. BMC Genomics 22, 52 (2021). https://doi.org/10.1186/s12864-021-07366-

7.There are many of these open-ended statements that are either meaningless in the broader context or without foundation. The authors are requested to revisit the text and to improve the quality (scientific language) and its readability so that others can at least attempt to corroborate what is being posited.

A: Thanks for your kind suggestions. We have carefully reviewed each sentence and made the necessary corrections.

Reviewer 2 Report

Comments and Suggestions for Authors

a nicely focussed paper  

good use of the tobacco system to probe roles of predicted effector genes

obviously  a huge number of other genes could be analyzed to indicate the range of mechanisms involved 

some sentences are unclear  

Comments on the Quality of English Language

generally  OK    some sentences though not clear    have caught some  but  likely other  examples exist 

Author Response

Reviewer2:

Line 21: Twenty-five

A: Thanks for your suggestion. We revised it as you requested.

2.Line 33: poor transition rewords

A: Thanks for your suggestion. We rewrote this part as you requested.

3.Line 39: effectors or effects?

A: Thank you very much for reminding us. We made a typo here. It should be "effectors" instead of "effects".

4.Line 42 Do not understand connections here.

A: Thank you very much. We rewrote this part in the manuscript and tried our best to make it clear.

5.Line 67 Unclear. Is this all of the secreted products or how is it determined which secreted products are those that are effectors.do not understand

A: Thank you very much. We rewrote this part in the manuscript and tried our best to make it clear.

  1. Line 74: could be better worded does not flow well

A: Thank you very much. We rewrote this part in the manuscript and tried our best to make it clear.

  1. Line 87:could be discussed more

A: Thank you very much. We rewrote this part as you requested. in the manuscript.

  1. line 120 ,139: Do not understand how old were the mycelia ?

A:Thanks for your question. We use 5-day-old mycelium here.

  1. line 227: what is mean by lesion area unclear?

A: Thanks for your question. In this study, “lesion area” refers to the leaf area where cell death occurs after the infection with V.mali and turns blue after trypan blue staining.

  1. line 296: in common with what please clarify?

A:Thanks for your question. We made a typo here. It should be “In all infection stages, 1919 proteins were found to be common in 1,2 and 5dpi.”

  1. fig 3C what are axes please label?

A: Thanks for your question. Usually, there is no X-axis in the upset plot, so we didn't label the X-axis. For more detailed information please refer to websites and papers blow.

https://r-graph-gallery.com/upset-plot.html

https://upset.app/

Duguet TB, Soichot J, Kuzyakiv R, Malmström L and Tritten L (2020) Extracellular Vesicle-Contained microRNA of C. elegans as a Tool to Decipher the Molecular Basis of Nematode Parasitism. Front. Cell. Infect. Microbiol. 10:217. doi: 10.3389/fcimb.2020.00217

  1. line314: A fungal effector is a

A: Thanks for your suggestion. We revised it as you requested.

  1. line 316:as part of their virulence.

A: Thanks for your suggestion. We revised it as you requested.

  1. line 364:to help readers may be a summary of such traits would be useful

A: Thanks for your suggestion. We rewrote this part as you requested.

  1. line 375,376:reword poor sentence structure.

A: Thanks for your suggestion. We rewrote this part as you requested.

16.line 390: have missed very few

A: Thanks for your suggestion. We revised it as you requested.

Round 2

Reviewer 1 Report

Comments and Suggestions for Authors

Please take a look at the recommendations in the file I've attached. The authors must derive a hypothesis from their introduction and formulate an AIM with a complete set of OBJECTIVES to fully inform the reader why this study was needed. And how the was to be carried out to answer - why it was needed. It is not acceptable to give a conclusion about what the study shows. Give the aim/objectives to inform what is being done, why and how. No more. Leave results for the Results and conclusions for the Conclusion.

And can we say that "the effector protein-coding genes" were unequivocally identified/characterised? If not, change to more appropriate wording - and throughout the MS.

Comments on the Quality of English Language

Writing style - grammar and typing need attention. Evidence of very little proof reading.

Author Response

1# Reviewer Comments

  1. Please take a look at the recommendations in the file I've attached. The authors must derive a hypothesis from their introduction and formulate an AIM with a complete set of OBJECTIVES to fully inform the reader why this study was needed. And how the was to be carried out to answer - why it was needed. It is not acceptable to give a conclusion about what the study shows. Give the aim/objectives to inform what is being done, why and how. No more. Leave results for the Results and conclusions for the Conclusion. And can we say that "the effector protein-coding genes" were unequivocally identified/characterized? If not, change to more appropriate wording - and throughout the MS. Line 97-107,This is NOT an AIM with a specific subset of OBJECTIVES. Please rewrite using the conventional scientific method approach.

A: Thank you very much for your review again, and also for your suggestions. We have revised it according to your suggestions.

  1. Line 114-116, Given that tissues have closely associated microbiomes, what is the purpose here?

A: Thanks for your question. In order to ensure the experimental results, we conducted all experiments under aseptic conditions. The surfaces of the newly collected twig and leaf samples carry many bacteria and impurities, which will affect the experimental results. Therefore, it is necessary to disinfect the surfaces with 75% ethanol before conducting the experiment.

  1. Line 117-118, Was this apparatus and the process under an aseptic technique? In any event, it seems a little contrived to ‘sterilize’ and then inoculate with another microbe and apparently not using aseptic technique.

A: Thanks for your question. Here we used fabric pattern wheel (As shown in Figure A blow) to puncture the leaves or the bark of tree branches. Firstly, we sterilize the fabric pattern wheel with an autoclave, and then gently roll it forward on the bark or leaves to puncture them (As shown in Figure B blow). Next, we put a mycelial plug on the wounds left by the fabric pattern wheel (As shown in Figure C&D blow). Fungi invade leaves and bark through these wounds. All inoculation experiments were completed under aseptic conditions in a laminar flow hood.

  1. We carefully revised the errors you indicated in Line 333,353,355 as you requested.
